# Kinetoplastid Species Maintained by a Small Mammal Community in the Pantanal Biome

**DOI:** 10.3390/pathogens11101205

**Published:** 2022-10-19

**Authors:** Filipe Martins Santos, Nayara Yoshie Sano, Sany Caroline Liberal, Maria Augusta Dario, Wesley Arruda Gimenes Nantes, Fernanda Moreira Alves, Alanderson Rodrigues da Silva, Carina Elisei De Oliveira, André Luiz Rodrigues Roque, Heitor Miraglia Herrera, Ana Maria Jansen

**Affiliations:** 1Programa de Pós-Graduação em Ciências Ambientais e Sustentabilidade Agropecuária, Universidade Católica Dom Bosco (UCDB), Campo Grande 79117-010, Brazil; 2Programa de Pós-Graduação em Ecologia e Conservação, Universidade Federal do Mato Grosso do Sul (UFMS), Campo Grande 79070-900, Brazil; 3Programa de Pós-Graduação em Biotecnologia, Universidade Católica Dom Bosco (UCDB), Campo Grande 79117-010, Brazil; 4Laboratório de Biologia de Tripanossomatídeos, Instituto Oswaldo Cruz (FIOCRUZ), Rio de Janeiro 21040-900, Brazil; 5Programa de Pós-Graduação em Biologia Parasitária, Instituto Oswaldo Cruz (FIOCRUZ), Rio de Janeiro 21040-900, Brazil

**Keywords:** wetland, Rodentia, Didelphimorphia, Next Generation Sequencing, Amplicon Sequence Variant

## Abstract

Kinetoplastids include species economically important in agriculture, livestock, and human health. We evaluated the richness of kinetoplastids that infect small mammals in patches of unflooded forests in the Pantanal biome, an area where we hypothesize that its diversity is higher than currently recognized. Hemocultures (HC) and Next Generation Sequencing (NGS) targeting the 18S rDNA gene were employed for the detection of kinetoplastids. We grouped the positive samples into pools for each small mammal species (*Monodelphis domestica*, *Thylamys macrurus*, *Oecomys mamorae*, *Thrichomys fosteri*, *Clyomys laticeps*, and *Holochilus chacarius*). Eight parasite species were identified: *Leishmania amazonensis*, *L. infantum*; *Trypanosoma cascavelli* (HC + NGS), *T. cruzi*, *T. lainsoni*, *T. rangeli* (HC + NGS), *Trypanosoma* sp. DID, and *Neobodo* sp. The use of a tool as sensitive as NGS has increased our awareness of the diversity of kinetoplastids, as well as their host range, with emphasis on the species *O. mamorae* (seven kinetoplastid species, excepting *T. cascavelli* in a pool of nine individuals) and *T. macrurus* (four kinetoplastid species in a single individual). Furthermore, *L. infantum* and *L. amazonensis* infections were described in small mammals from this region for the first time. These findings make it mandatory to revisit the kinetoplastids/host associations proposed so far.

## 1. Introduction

The class Kinetoplastea are protists belonging to the phylum Euglenozoa [1] and are characterized by the presence of a kinetoplast, a large mass of mitochondrial DNA considered an apomorphy of the group (kDNA) [2]. Different species of kinetoplastids are classified not only by their morphological characteristics but also by their disease manifestation and life cycle, such as free-living, monoxenous, or heteroxenous, and include both intracellular and extracellular stages [3,4]. In this sense, organisms are traditionally subdivided into two groups: parasitic trypanosomatids and mostly free-living bodonids, leaving the latter as a paraphyletic group, and phylogenetic relationships both between and within the main groups of kinetoplastids are still not reliably resolved [5,6].

Due to their importance for human and animal health, trypanosomatids (family Trypanosomatidae), a group characterized by having a single flagellum, have been studied much more intensively than the other protozoa from this class. This family is composed of obligate parasites capable of infecting invertebrates, vertebrates, and plant hosts and includes species responsible for economic losses in agriculture and livestock, as well as important human diseases [7,8]. Within this family, nineteen genera are recognized as monoxenous, and six genera are classified as heteroxenous trypanosomatids [7,9,10,11,12]. Within this second group, we highlighted *Leishmania* and *Trypanosoma*, maintained in natural and anthropic environments in complex transmission cycles by a wide diversity of vertebrate animal species in the Americas [13].

Species belonging to *Trypanosoma* have been recorded in the Nhecolandia subregion of Pantanal for the last two decades parasitizing several species of wild mammals and invertebrate hosts [14,15,16,17,18,19,20]. These studies indicate that the maintenance of trypanosome species includes several species of wild mammals with different infective competencies in complex, unpredictable, and multivariate wild cycles linked to the various interactions between host species and land use [14,15,20]. Despite this effort, the diversity of Kinetoplastea may be higher in the region since Santos et al. [20] observed that some uncharacterized flagellates, detected by hemoculture from three species of small mammals (*Clyomys laticeps*, *Cerradomys scotti* and *Monodelphis domestica*), were unable to spontaneously grow and be isolated.

Currently, the most common molecular method for diagnosing kinetoplastid infection is the Sanger sequencing of amplicons resulting from positive PCRs targeting conserved genes of these parasites, mainly 18S rDNA and gGAPDH [20,21,22]. Although accurate, fast, and sensitive, this technique is not effective in the detection of co-infections [23,24,25]. The use of different species-specific PCR assays was proposed to resolve co-infections within trypanosomes; however, these tools are limited to known species that circulate in the study area, which may limit the possibility of noticing new or rare species [20,25,26,27,28]. In contrast, next-generation sequencing (NGS) presents higher efficiency, allowing the identification of less abundant genotypes/species overlooked by Sanger sequencing, which is essential in understanding the ecology of these parasites in communities with a high diversity of wild hosts and parasites [21]. This technique generates thousands of sequences from a single sample resulting in a secure base of reads that allows us to predicate the diversity of organisms that infect a given host [29,30]. By using the NGS technique, we hypothesized that it would be possible to detect a higher richness of kinetoplastid protists infecting small mammal species in patches of unflooded forests from the Pantanal region than previously recognized.

## 2. Materials and Methods

### 2.1. Study Area

The study was conducted in a private ranch in the center of Brazilian Pantanal, Nhecolândia sub-region, in Corumbá municipality (Figure 1). Pantanal is unique in geomorphological formations that can be divided into 11 sub-regions [31], among which is the Nhecolândia, an area characterized by a mosaic of vegetation composed mainly of a seasonally flooded grassland and patches of semi-deciduous forest, with some shrub vegetation. The forested area is located at higher elevations, referred to as “cordilheiras” (long strings of forest), that remain unflooded during the rainy season. The ground is composed mainly of leaf litter, and the canopy is discontinuous, reaching about 5–6 m in the central zones [32]. The understory is composed mainly of spiny bromeliads (*Bromelia balansae*) on the edges and Acuri palms (*Attalea phalerata*) inside, which attain heights of approximately 1–2 m [33]. The climate is marked by two distinct seasons: the rainy season (October to March) and the dry season (April to September).

### 2.2. Captures and Blood Sampling

Small mammals were captured inside 10 forest patches in February 2021. We used 2040 trap nights (204 H. B. Sherman Traps, Tallahassee, FL, USA, and 204 Tomahawk Live Traps, Tomahawk, WI, USA) distributed according to the bromeliad cover. Traps were baited daily with a mixture of banana, peanut butter, and cornmeal.

The individuals captured were chemically anesthetized using a combination of ketamine hydrochloride (100 mg/mL) and acepromazine (10 mg/mL) (9:1) intramuscularly. They were euthanized by exsanguination during cardiac puncture, and part of the blood was used for hemoculture (HC). Thereafter, spleen and liver fragments were collected under sterile conditions, and the remaining blood was separated into blood clots (employed in molecular assays) and serum. All methods used in this study were approved by the Ethics in Animal Research Committee of the Dom Bosco Catholic University under the permission no. 013/2020, and this study was conducted under the authorization of Chico Mendes Institute of Biodiversity and Conservation (no. #70946-4).

### 2.3. Kinetoplastid Detection

The HCs were performed in duplicate using 200 to 300 µL of blood, which were inoculated in individual tubes containing NNN/LIT medium (Novy–MacNeal–Nicolle/liver infusion tryptose). The cultures were examined every two weeks for at least two months. When parasites isolated in the positive HCs reached the exponential phase, they were subjected to DNA extraction using the phenol–chloroform method described by [34] and deposited in the *Coleção de Trypanosoma de Mamíferos Silvestres*, *Domésticos e Vetores*, *Fiocruz* -COLTRYP (*Oswaldo Cruz Foundation*, *Rio de Janeiro*-RJ/Brazil, available at www.coltryp.fiocruz.br (accessed on 22 March 2022)). Positive cultures that did not result in parasite amplification for isolation (cultures not established) were directly centrifuged, and the sediments were stored at −20 °C until DNA extraction, according to [35].

The DNAs derived from the culture were subjected to the nested Polymerase Chain Reaction (nPCR) targeting the kinetoplastid 18S rDNA gene (~650 bp) [36]. Products of nPCR were purified using the Illustra GFX PCR DNA and gel band purification kit (GE Healthcare Life Sciences, Little Chalfont, Buckinghamshire, UK). All samples were sequenced for both strands of DNA with BigDye Terminator v3.1 Cycle Sequencing Kit (Applied Biosystems, Foster City, CA, USA) on an ABI 3730 DNA sequence available on the PDTIS/FIOCRUZ sequencing platform. The sequences were edited, aligned, and corrected using BioEdit software.

We also performed nPCR for the molecular diagnosis of kinetoplastid detection in DNA samples derived from the blood clots, spleen, and liver of sampled hosts [36]. Genomic DNA of clots was extracted according to Rodrigues et al. [22], and the spleen and liver were extracted using the QIAamp Blood DNA Mini Kit (Qiagen, Hilden, Germany) according to the manufacturer’s instructions. Any individual that was positive in PCR in any of the three samples used (blood clots, spleen, and/or liver) was considered infected by kinetoplastids. We grouped the positive samples of the 18S rDNA gene into a pool according to the small mammal species. These samples were prepared for NGS according to Illumina recommended protocols (Illumina Demonstrated Protocol: Metagenomic Sequencing Library Preparation) and sequenced on an Illumina HiSeq2500 (PE250) following primers described by Barbosa et al. [21].

### 2.4. Bioinformatics Analysis

The NGS-generated data were imported into the R v3.6.2 environment, and wherein all the bioinformatic analysis was carried out [37]. The sequences were analyzed using the DADA2 package v1.14.0 following the tutorial (https://benjjneb.github.io/dada2/tutorial.html (accessed on 20 April 2022)) [38]. Further, taxonomy was assigned using SILVA v132. The Amplicon Sequence Variant (ASV) table, assigned taxonomy, and sample metadata information were combined as a phyloseq object (phyloseq package version 1.30.0) [39]. As a read cut-off for determining the species occurrence per sample, the total reads per sample obtained in the ASV table were normalized to 100,000 reads, and AVS that presented ≤50 reads in the sample were excluded from the analysis [29].

### 2.5. Phylogenetic Analysis

For kinetoplastid species/genotype identification and genetic clustering, the ASV reads were aligned to kinetoplastid 18S rDNA species sequences retrieved from the GenBank database in MAFFT v.7.0 [40] using the L-INS-i algorithm. The alignment was visualized and manually edited on MegaX [41]. Maximum likelihood (ML) estimation and Bayesian inference (BI) were performed as follows: for each phylogenetic analysis, the best base substitution models were chosen according to the corrected Akaike information criterion (cAIC) in JmodelTest 2.1.10 v20160303 [42]; ML reconstruction was performed in the IQ-Tree program [43] on PhyloSuite v.1.2.2 [44] using Ultrafast bootstrapping [45] with 5000 replicates, 1000 maximum interactions and 0.99 minimum correlation coefficients for branch support. The SH-aLRT branch test with 5000 replicates was also applied to validate the Ultrafast bootstrap; for BI reconstruction, the Bayesian–Markov chain Monte Carlo (MCMC) method was performed in Bayesian Evolutionary Analysis Sampling Trees (BEAST) v2.6.2 [46]. Eight independent runs were performed for 20 M with sampling every 2000 generations and pre-burn in of 5 M. The birth–death and Yule model specifications were used in tree reconstructions. The parameters selected that led to effective sample size (ESS) higher than 500 were considered appropriate and were observed in TRACER v.1.6 [47]. Three to four runs converged were calculated after 25% of each run was excluded (burn-in) in LogCombiner. The final tree was generated with maximum clade credibility (MCC) after 25% burn-in and a 0.6 posterior probability limit (PP) in Tree Annotator. Both the ML and BI reconstructions were visualized in Figtree v.1.4.3, and a final tree was generated.

For *Trypanosoma cascavelli* intra-specificity, a haplotype network was constructed by median-joining [48] and maximum parsimony [49] post-processed clean-up procedure available on Network software version 4.0.1.1 (fluxus-engineering.com (accessed on 25 April 2022)).

## 3. Results

We captured 33 individuals belonging to six species. Two of these species of didelphids were marsupials, *Monodelphis domestica* (*n* = 04) and *Thylamys macrurus* (*n* = 01), and four species were rodents, *Oecomys mamorae* (*n* = 12), *Thrichomys fosteri* (*n* = 10), *Clyomys laticeps* (*n* = 05), and *Holochilus chacarius* (*n* = 01). Nine HCs (09/33) were positive for flagellate’s forms, but the characterization was possible for four samples: one *Trypanosoma rangeli* A (LBT 12592; Coltryp 912), isolated from *T. fosteri*; one *T. rangeli* B (LBT 12613; Coltryp 914), isolated from *O. mamorae*; and two *Trypanosoma cascavelli,* characterized by sediments (LBT 12589 and LBT 12598) from *M. domestica*. We observed that 76% (25/33) of collected individuals were positive for kinetoplastids by 18S rDNA (*M. domestica* [03/04], *T. macrurus* [01/01], *O. mamorae* [09/12], *T. fosteri* [09/10], *C. laticeps* [02/05] and *Holochilus chacarius* [01/01]).

A summary of the NGS sequencing data quality is shown in Appendix A. A total of 417,293 raw sequences were obtained. Then, 355,179 sequences were selected after the preliminary quality filtering. Denoised F/R quality filtering was then performed, which yielded 354,035 (denoised F) and 354,133 (denoised R) sequences. The total number of merged forward-reverse reads was 350,587 sequences, and 344,885 sequences were selected for analysis after chimera removal. Finally, the relative number of passed reads after all the above steps ranged from 93.89 to 100%.

In the 436 sequences that constituted the database, we observed ten groups of kinetoplastids, according to the phylogenetic analysis: *Trypanosoma cruzi* Discrete Typing Unit (DTU) TcI (ASV1 and ASV2), *T. cruzi* DTU TcII (ASV3), *Trypanosoma rangeli* lineages A (ASV12), B (ASV4) and E (ASV11) and *Trypanosoma* sp. DID (ASV5) (Figure 2); *Trypanosoma lainsoni* (ASV7) and *T. cascavelli* (ASV6) (Figure 3); *Leishmania amazonensis* (ASV8); *L. infantum* (ASV9) (Figure 4); and *Neobodo* spp. (ASV10) (Figure 5).

The small mammal species that showed higher kinetoplastid richness was *O. mamorae* with all kinetoplastid species, except for *T. cascavelli*, followed by *T. macrurus*, *C. laticeps*, *M. domestica*, and *T. fosteri*; and *Holochilus chacarius* (Table 1). *Leishmania infantum* was found in all small mammal species, and *L. amazonensis* was present in three species (*O. mamorae*, *T. fosteri*, and *C. laticeps*). Furthermore, we detected *Neobodo* spp. only in *O. mamorae* and *T. cascavelli* only in *M. domestica*. The phylogenetic analysis of two *T. cascavelli* hemocultures and the NGS results (Figure 3 and Figure 6) revealed that this trypanosomatid species could be divided into two groups: one exclusively occurring in marsupials (Figure 6) and another associated with both reptiles and mammals.

**Table 1 pathogens-11-01205-t001:** Trypanosomatidae species detected in Pantanal’s small mammals. The column names are referred to each small mammal species evaluated, such as Omam: *Oecomys mamorae*; Tfos: *Thrichomys fosteri*; Clat: *Clyomys laticeps*; Mdom: *Monodelphis domestica*; Hcha: *Holochilus chacarius*; Tmac: *Thylamys macrurus*. In parenthesis, we indicate the number of positive/total individuals in the 18SrDNA PCR. The dark blue color represents positive small mammal species, and the light blue color represents negative species.

Species	Mdom (3/4)	Tmac (1/1)	Omam (9/12)	Tfos (9/10)	Clat (2/5)	Hcha (1/1)
*Leishmania amazonensis* (ASV8)						
*Leishmania infantum* (ASV9)						
*Trypanosoma cascavelli* (ASV6)						
*Trypanosoma cruzi* DTU TcI (ASV1)						
*Trypanosoma cruzi* DTU TcI (ASV2)						
*Trypanosoma cruzi* DTU TcII (ASV3)						
*Trypanosoma lainsoni* (ASV7)						
*Trypanosoma rangeli* A (ASV11)						
*Trypanosoma rangeli* B (ASV4)						
*Trypanosoma rangeli* E (ASV12)						
*Trypanosoma* sp. DID (ASV5)						

The pool of each small mammal consisted of the following: *Oecomys mamorae*: blood C = clot + liver + spleen; *Thrichomys fosteri*: blood clot + liver + spleen; *Clyomys laticeps*: liver + spleen; *Monodelphis domestica*: blood clot + liver + spleen; *Holochilus chacarius*: liver; *Thylamys macrurus*: blood clot + liver.

**Figure 6 pathogens-11-01205-f006:**
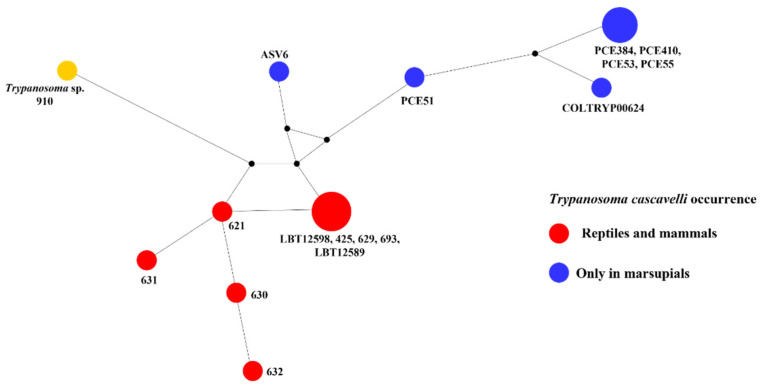
*Trypanosoma cascavelli* haplotype network. Networks were constructed with 18 SSU rDNA sequences, and the size of each node is proportional to the sequence frequency. The small black circle represents the median vector, which can be interpreted as an unsampled or an extinct ancestral sequence. The red dots represent *T. cascavelli* occurrence in reptiles and marsupials; the blue dots represent *T. cascavelli* occurrence only in marsupials; the yellow dot represents *Trypanosoma* sp. (EU095838) closely related to snake trypanosomatids that occurred in sandfly. The *T. cascavelli* sequences used in the analysis were: PCE51 (MH411650); PCE53 (MH411651); PCE55 (MH411652); PCE384 (MH411653); PCE410 (MH411656); COLTRYP00624 (MF141867); 693 (EU095845); 632 (EU095844); 631 (EU095843); 630 (EU095842); 629 (EU095841); 621 (EU095840); 425 (EU095837); LBT12589 (OP508234); LBT12598 (OP508235).

## 4. Discussion

Our results showed that NGS technologies could be used to identify kinetoplastid species that were not recorded previously by the usual diagnostic techniques in small mammals from the Nhecolândia sub-region, increasing the knowledge about the richness of kinetoplastid in the region. Despite the inherent difficulties, the study of small mammals is fundamental to improving the understanding of kinetoplastid ecology. These groups of hosts are found both on the ground and in the canopy and may favor the dispersion of parasites between the strata of the forest environment [20,32]. Furthermore, small mammal species are abundant in the Pantanal region and participate in the base of trophic transmission, consequently assuring the maintenance of trypanosomatid species in the environment by predation (oral route), for parasites known to be transmitted by this route. Indeed, the sharing of habitats among host species could enhance parasite spillover [20]. Our findings reinforce the complexity of the Kinetoplastea ecology and advert oversimplified interpretations of the associations between these clades and host species. Ecological parasite adaptation, involving the exchange of hosts within the same ecological niche, has been considered in Kinetoplastea evolution [50]. With the improvement of molecular techniques, there are increasing reports of parasitic species infecting unusual hosts. For instance, *T. cascavelli*, originally described in snakes, has already been recorded in opossums [50]; *T. lainsoni*, described previously in rodents, has been found in marsupials and bats; and *T. dionisii*, classically associated with bats, was found in humans and marsupials [22,30,51,52].

Despite the fact that *T. cascavelli* has already been found by molecular test in *Monodelphis americana* [53], *Didelphis albiventris*, and *Marmosa demerarae* [22], we observed this parasite in HCs in two of four *M. domestica* examined (although these cultures were not established). The presence of *T. cascavelli* in HCs may indicate that *M. domestica* has the potential to be a source of infection to its (still unknown) vector. The infection of these marsupial host species may be occurring through sand flies [54], which were widespread in the studied area [55,56], and was implicated as putative vectors of this *Trypanosoma* species [54]. Additionally, *T. cascavelli* infecting *M. domestica* in the Pantanal biome increased the knowledge about the geographic distribution of this trypanosomatid species, previously restricted to the Atlantic Forest biome. Moreover, our results showed the ability of *T. cascavelli* to parasitize another species of *Monodelphis* since previous reports were related to *M. americana*. Our data support the hypothesis that marsupial species may be the ancestral hosts of *T. cascavelli* and snakes are accidental hosts, which may be infected through the predation of small mammals [53].

All six pool samples sent to NGS analysis were constituted for liver samples. This could allow us to detect *L. amazonensis* and *L. infantum* in small mammal species in South Pantanal for the first time. The use of parasitological tests from whole blood could mask the records of *Leishmania* spp. infecting hosts in this region due to the specialization in parasitic cells of the mononuclear phagocytic system, substantially decreasing the sensitivity of parasitological tests. We detected *L. infantum* in all small mammal species sampled, suggesting its wide distribution in the region, while *L. amazonesis* were found only in the rodents *Oecomys mamorae*, *T. fosteri*, and *C. laticeps*. This divergence may be related to their microhabitat, as these mammal species use spiny bromeliad (*B. balansae*) as shelter, a favoring place for vectors, such as *Lutzomyia cruzi* [57]. Oppositely, *M. domestica* and *T. macrurus* are usually found within forests, and the semi-aquatic rodent *Holochilus chacarius* is usually found in flooded areas [32,33].

Although several studies have shown the presence of *Leishmania* in the neighborhood region of Corumbá, an endemic area for leishmaniasis, *L. braziliensis* was recently reported by molecular techniques in phyllostomid bats in the studied area [58]. The several mammal species found in the present study with *L. amazonensis* and *L. infantum*, together with the presence of their vector *Lu. cruzi* in the same studied area-permissive vector of *L. amazonensis* [57,59,60,61] shows a possible, stable transmission cycle of *Leishmania* spp. in the South Pantanal. The NGS technique allowed us to detect infections by *L. infantum* and *L. amazonesis* in three rodent species. Indeed, studies carried out in humans previously diagnosed with *L. infantum* showed through NGS the existence of co-infections by two or more species, including *L. infantum*, *L. amazonensis*, *L. braziliensis*, *L. panamensis*, *L. lindenbergi*, *L. naiffi*, and *T. cruzi* [62], showing the high sensitivity of this technique.

Regarding *T. rangeli*, we observed three of the five lineages currently described [63,64]. The three lineages detected in the present study have already been described as infecting two species of Carnivora *Nasua nasua* (lineages A and B) and *Procyon cancrivorus* (lineage E), and Cingulata *Priodontes maximus* (lineage E) in the Pantanal [65]. *Trypanosoma rangeli* has been described as exclusively transmitted by triatomines of the genus *Rhodnius*, which has been widely described as colonizing *N. nasua* arboreal nests in this region [19,20,66]. Since nests of *N. nasua* can act as a hub of many mammals species, such as the small rodents *O. mamorae* and *T. fosteri* [20], as well as the arboreal species *T. macrurus* and *Tamandua tetradactyla* (Herrera personal comm), maybe the lineages A and B would be occurring only in understory strata.

We also detected the kinetoplastid *Neobodo* spp. These heterotrophic flagellates are generally found in the aquatic environment and have not yet been associated with parasitism [67]. Indeed, bodonids belong to the most frequent protist groups in some marine communities, and the most possible explanation for their presence in small mammals is the acquisition by contaminated water. Although bodonid DNA has been reported in samples from vertebrate and invertebrate hosts on previous occasions [30,68,69], finding DNA in mammalian samples does not necessarily indicate an infection in tissues. It is possible that genes can prevent the degradation of the digestive tract through an unknown mechanism entering the circulation system [70].

Our results point out that *Trypanosoma* sp. DID may parasitize a wider range of marsupial species than previously reported. With our data, we added the observation of *Trypanosoma* sp. DID in *T. macrurus*, which was previously reported only in *Didelphis albivestris* and *D. aurita*. Despite being described as infecting didelphids [22,51,52], we also detected *Trypanosoma* sp. DID in two rodent species for the first time, the scansorial *Oecomys mamorae* and the semifossorial *C. laticeps*. As reported by Dario et al. [29], we did not observe any phylogenetic difference between the sequences detected here and those previously detected.

Small mammals constitute one of the most abundant prey for carnivorous species in the Pantanal biome, reinforcing the importance of this group for the food web [71]. As predation has been suggested as a source of infection by *T. cruzi* and *T. evansi* for carnivores [14], the small mammal community of the Pantanal region may also be playing an important role in the enzooty of other trypanosomatids, such as *L. infantum*, *L. amazonensis*, *T. cascavelli*, *T. lainsoni*, and *T. rangeli*, ensuring oral infection among free-living carnivores, including top endangered predators. It is worth mentioning that oral infection is proposed to occur in different trypanosomatid parasites and was recently demonstrated to occur also for *L. infantum* and *L. braziliensis* [72]. The use of a sensitive tool such as the NGS has increased our knowledge about kinetoplastid diversity, making it possible to identify up to seven species (*L. amazonensis*, *L. infantum*, *T. cruzi*, *T. lainsoni*, *T. rangeli*, *Trypanosoma* sp. DID and *Neobodo* spp.) in an *O. mamorae* pool samples and allowed us to find in a single individual of a mouse–opossum *T. macrucus*, a coinfection of *L. infantum*, *T. cruzi* DTU (TcI and TcII), *T. rangeli* B and *Trypanosoma* sp. DID, species that, until the present moment, had no report of kinetoplastid infection. Our findings make it mandatory to revisit the kinetoplastid/host associations proposed so far.

## Figures and Tables

**Figure 1 pathogens-11-01205-f001:**
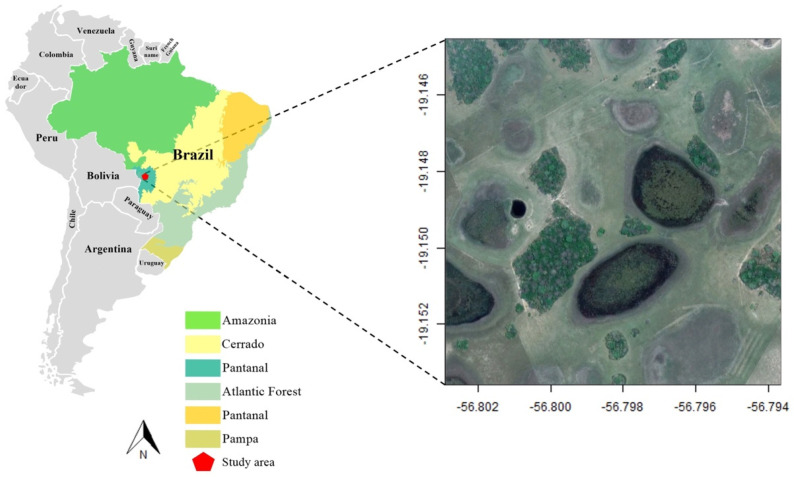
Map of the study area. On the left is the map of South America, highlighting the Brazilian Biomes in colors. In the right is the forest patches where small mammal samples were collected in the present study.

**Figure 2 pathogens-11-01205-f002:**
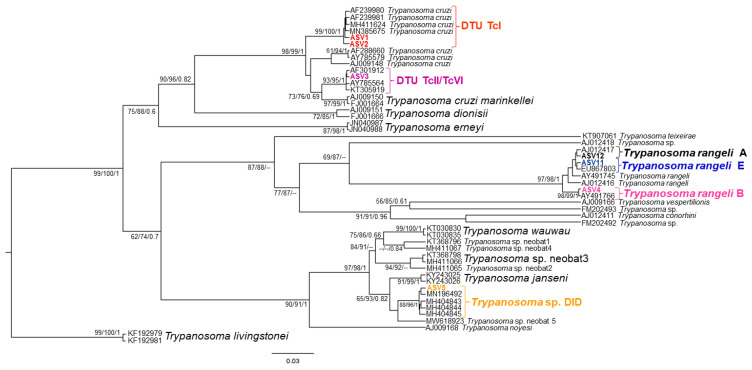
*Trypanosoma cruzi* clade phylogenetic tree based on 18S rDNA gene for liver, spleen, and blood clot DNA pool from different small mammal species. The trees were inferred with the transition model plus gamma distribution among sites (TIM3 + G). The number of nodes correspond to ML (ultrabootstrap and SH-aLRT) and BI (posterior probability). The scale bar shows the number of nucleotide substitutions per site. The red bracket indicates the group formed by *T. cruzi* DTU TcI; the lilac bracket indicates the sequences grouped as *T. cruzi* DTU TcII/TcVI; the black, blue, and pink brackets indicate the sequences identified as *T. rangeli* lineages A, E, and B, respectively; and the yellow bracket indicates the sequences identified as *Trypanosoma* sp. DID.

**Figure 3 pathogens-11-01205-f003:**
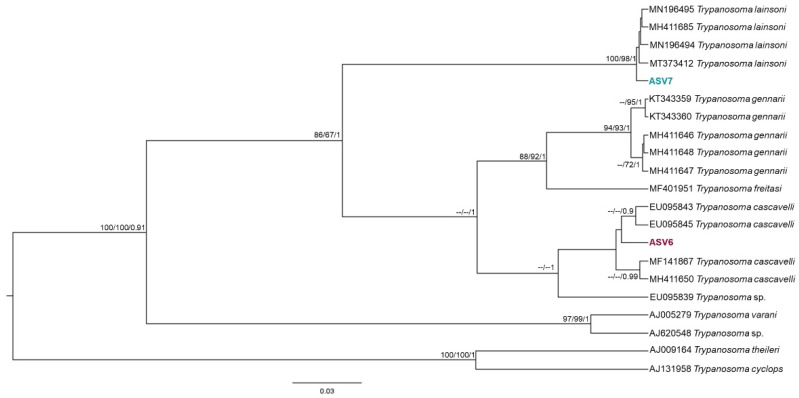
Trypanosomatid of reptile and mammals’ clade phylogenetic tree based on 18S rDNA gene for liver, spleen, and blood clot DNA pool from different small rodent species. The phylogenetic tree was inferred with the transition model plus gamma distribution among sites (TIM3 + G). The number of nodes correspond to ML (ultrabootstrap and SH-aLRT) and BI (posterior probability). The scale bar shows the number of nucleotide substitutions per site. The ASV6 grouped together with other *T. cascavelli* sequences from marsupial species. The ASV7 grouped with *T. lainsoni* sequences from different mammal species and biomes.

**Figure 4 pathogens-11-01205-f004:**
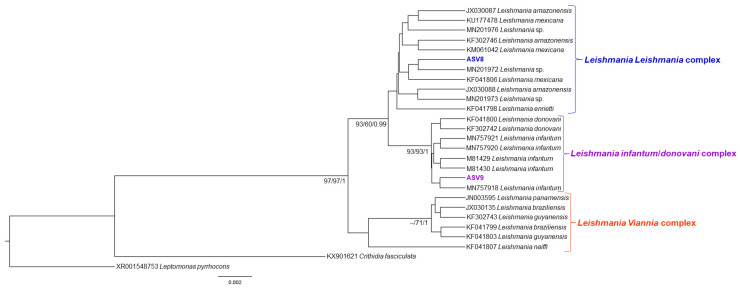
*Leishmania* spp. phylogenetic tree based on 18S rDNA gene for liver, spleen, and blood clot DNA pool from different small mammal species. Phylogenet tree was inferred with the transition model plus gamma distribution among sites (TIM3 + G). The number of nodes correspond to ML (ultrabootstrap and SH-aLRT) and BI (posterior probability). The scale bar shows the number of nucleotide substitutions per site. The blue, purple, and red brackets correspond to the *Leishmania* (*Leishmania*), *L. infantum*/*donovani*, and *L.* (*Viannia*) complexes, respectively. The ASV8 grouped with *Leishmania* sequences from the *Leishmania* (*Leishmania*) complex, and the ASV9 grouped with *Leishmania* sequences from the *Leishmania infantum/donovani* complex.

**Figure 5 pathogens-11-01205-f005:**
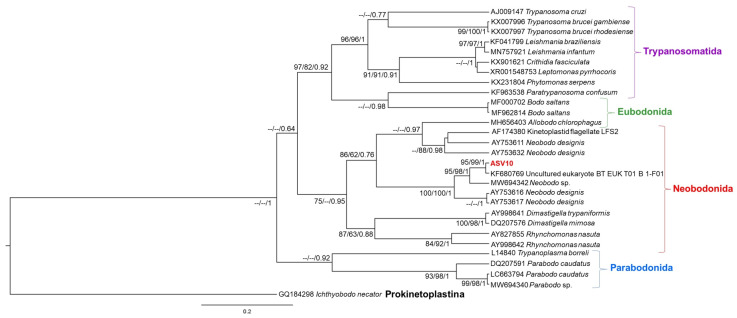
Kinetoplastea class phylogenetic tree based on 18S rDNA gene for liver, spleen, and blood clot DNA pool from different small mammal species. Phylogenet tree was inferred with the PM1 uf model plus gamma distribution among sites (TPM1 uf + G). The number of nodes correspond to ML (ultrabootstrap and SH-aLRT) and BI (posterior probability). The brackets’ colors correspond to the following trypanosomatid orders: purple—Trypanosomatida; green—Eubodonida; red—Neobonida; blue—Parabodonida. The ASV10 grouped together with *Neobodo* sequences from the Neobonida order.

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
