# Peer review of "Kinetoplastid Species Maintained by a Small Mammal Community in the Pantanal Biome"

_pathogens, 2022, doi:10.3390/pathogens11101205_

Round 1

Reviewer 1 Report

Santos et al use next generation sequencing approaches to show that the diversity of kinetoplastid species detected in a range of small mammals from the Pantanal region of Brazil is larger than previously estimated. Overall the manuscript is readable and the discussion is nicely put together. However, I'm uncertain of the actual value of the manuscript since in many ways it is no surprise that applying a sensitive sequencing technique detects a broader range of kinetoplastid species in forest mammals than previously thought.

There are only minor revisions that I would suggest:

1. a map showing the field site area for sampling would be appropriate and of interest to readers from outside of Brazil

Line 24 'is higher' not 'in higher'

Line 83 'hypothesised' not 'hypothesise' - care needed to ensure the past tense is maintained throughout the manuscript.

Author Response

Reviewer 1:

Comments and Suggestions for Authors

Santos et al use next generation sequencing approaches to show that the diversity of kinetoplastid species detected in a range of small mammals from the Pantanal region of Brazil is larger than previously estimated. Overall the manuscript is readable and the discussion is nicely put together. However, I'm uncertain of the actual value of the manuscript since in many ways it is no surprise that applying a sensitive sequencing technique detects a broader range of kinetoplastid species in forest mammals than previously thought.

Repost Reviewer 1: Thanks for the reviews and all suggestions were accepted. One of the most important points of the article was that through the approach of the next generation sequencing, it was possible to find unusual trypanosomatid species, probably because they do not show enough parasitemias necessary for detection by molecular tools previously used. We report here for the first time Leishmania infantum at pantanal floodplain, as well as Leishmania amazonensis in small mammals. In addition, we increased the geographic distribution of Trypanosoma cascavelli, which was previously described only in the Atlantic Forest region. The host range also increases as Trypanosoma sp. DID, although not fully understood, was first found in the marsupial Thylamys macrurus and in the two rodent species Oecomys mamorae and Clyomys laticeps.

Reviewer 1_1: 1. a map showing the field site area for sampling would be appropriate and of interest to readers from outside of Brazil.

Repost Reviewer 1_1: We appreciate the suggestion and we've added the map – Line 105.

Reviewer 1_2: Line 24 'is higher' not 'in higher'.

Repost Reviewer 1_2: We accepted the suggestion.

Reviewer 1_3: Line 83 'hypothesised' not 'hypothesise' - care needed to ensure the past tense is maintained throughout the manuscript.

Repost Reviewer 1_3: We accepted the suggestion.

Reviewer 2 Report

Generally speaking, this manuscript by Santos FM and colleagues is well written, and it is suitable for publication is this journal.  The following are a few minor points:

L24: “its diversity is higher”

L45: change lifestyle to life cycle.

L50-1: “(family Trypanosomatidae)”

References: The first letter of a species should not be capitalized although it is required for a genus.

Author Response

Reviewer 2

Comments and Suggestions for Authors

Generally speaking, this manuscript by Santos FM and colleagues is well written, and it is suitable for publication is this journal.  The following are a few minor points:

Repost Reviewer 2: Thanks for the reviews and all suggestions were accepted.

Reviewer 2_1: L24: “its diversity is higher”

Repost Reviewer 2_1: We accepted the suggestion.

Reviewer 2_2: L45: change lifestyle to life cycle.

Repost Reviewer 2_2: We accepted the suggestion.

Reviewer 2_3: L50-1: “(family Trypanosomatidae)”

Repost Reviewer 2_3: We accepted the suggestion.

Reviewer 2_4: References: The first letter of a species should not be capitalized although it is required for a genus.

Repost Reviewer 2_4: We regret the mistake and correct them all.